# Investigation of Dynamic Behavior and Process Stability at Turning of Thin-Walled Tubular Workpieces Made of 42CrMo4 Steel Alloy

Kamel Mehdi [1,2,*], Peter Pavol Monka [3,4], Katarina Monkova [3,4,*], Zied Sahraoui [2], Nawel Glaa [2] and Jakub Kascak [3]

1 Preparatory Institute for Engineering Studies El Manar (IPEIEM), University of Tunis EL Manar (UTM), PB. 244, Tunis 2092, Tunisia
2 Production and Energetics Laboratory (LMPE), Engineering National High School of Tunis (ENSIT), Mechanics, University of Tunis (UT), 5 Avenue Taha Hussein, P.B. 56, Bab Mnara, Tunis 1008, Tunisia; zied.sahraoui@outlook.com (Z.S.); nawelglaa2020@gmail.com (N.G.)
3 Faculty of Manufacturing Technologies, Technical University in Kosice, Sturova 31, 080 01 Presov, Slovakia; peter.pavol.monka@tuke.sk (P.P.M.); jakub.kascak@tuke.sk (J.K.)
4 Faculty of Technology, Tomas Bata University in Zlin, Nam. T.G. Masaryka 275, 760 01 Zlin, Czech Republic
* Correspondence: kamel.mehdi@ipeiem.utm.tn (K.M.); katarina.monkova@tuke.sk (K.M.)

**Abstract:** During machining, the surface of the machined materials is damaged and tool wear occurs, sometimes even to complete failure. Machining of thin-walled parts is generally cumbersome due to their low structural rigidity. The study deals with the effect of the feed rate and the thickness of the thin-walled part on the dynamic behavior and stability of the turning process during the roughing and finishing of thin-walled tubular workpieces made of steel alloy 42CrMo4. At the same time, the cutting forces and deformations of the workpiece were also evaluated via numerical and experimental approaches. The numerical study is based on a three-dimensional (3D) finite element model (FEM) developed using the ABAQUS/Explicit frame. In the model, the workpiece material is governed by the behavior law of Johnson–Cook. Numerical and experimental results show that the cutting forces and the quality of the machined surface depend not only on the choice of cutting parameters but also on the dynamic behavior of thin-walled parts due to their low rigidity and low structural damping during the machining operation. Cutting forces are proportional to the feed rate and inversely proportional to the thickness of the part. Their variations around the average values are low for roughing tests where the wall-part thickness is higher or equal to 3.5 mm. However, these variations intensify for finishing tests where the wall thickness is less or equal to 1.5 mm. Indeed, the recorded FFT spectra for a finishing operation show several harmonics that occurred at around 550 Hz, and the amplitude of the peaks, which describes the level of power contained in the signals, shows an increase similar to that of the amplitudes of the temporal signal. The flexibility of the part generates instability in the cutting process, but the frequencies of the vibrations are higher than the frequency of rotation of the part.

**Keywords:** machining; turning; thin-walled tubular workpieces; stability; 3DFEM; cutting forces; vibrations; lobes diagrams; simulation





## 1. Introduction

In the machining field using cutting tools, the cutting process investigations and their optimization are still very delicate and financially costly. This is mainly due to the complexity of the multi-physical phenomena governing the tool–matter interaction [1]. Thus, recourse to the use of mathematical modeling tools for the prediction of cutting tool/matter behavior becomes essential [2,3]. These prediction tools are closely linked to the material properties of the machined parts as well as to the phenomena that accompany

the cutting tool and the workpiece interaction. The modeling tools are only relevant if they are rigorously constructed and fed with reliable experimental data reflecting the reality of this cutting process [4,5]. The models generally developed in the literature have as objective functions the cutting phenomena (vibrations, chatter, temperature, forces, wear, etc.) where the characteristics of the machined part are linked to its integrity and its resistance to fatigue (roughness, residual stresses, topography, hardness, microstructure, etc.) [6–10]. Given the diversity of the results offered by these simulation models as well as the low implementation costs compared to those relating to experimental studies, many researchers have been interested in the development of finite element models (FEM) in order to study the effect of cutting parameters on the machining process and predict chip formation and its morphology [11]. Likewise, other models have been developed to predict cutting forces and heat transfers during machining [12–15] and study the influence of these parameters on the chip morphology [16–18]. Pantale [19] performed numerical simulations of continuous and discontinuous chip formation during the cutting process of the 42CrMo4 material in 2D and 3D based on the orthogonal cutting model using ABAQUS/Explicit dynamics. The model is based on the combined formulation ALE (arbitrary Lagrangian–Eulerian) for quadrilateral elements with four nodes for the model in 2D and brick elements with eight nodes for the model in 3D. The Johnson–Cook law is adopted, taking into account the effects of the strain rate and cutting temperature. The classical contact Coulomb law is adopted with a friction coefficient value µ equal to 0.32. In the simulation model, the cutting tool is considered a rigid body, and the workpiece is a deformable part. The results of the simulation described the distribution of von Mises stress during chip formation as well as the variation in cutting force. Chip formation during the turning process of aluminum alloy A2024-T351 was studied numerically by Mabrouki [20] using a FEM. In this study, an innovation was introduced in the model, which consists of the coupling between the Johnson–Cook damage model and the fracture energy of the material of the workpiece. The numerical results showed the presence of small radii of curvature on the formed chip, and the fragmentation is the result of the bending forces caused both by the effect of the tool feed rate and the surface contact between the chip and the workpiece. Yue [21] simulated the cutting of hard steel GCr15 to predict the variation of the cutting forces and the morphology of the chip by developing a 3D FEM using the ABAQUS/Explicit in which the Johnson–Cook law has been employed to model the behavior of the machined material. To minimize the computation time, a local refinement was introduced in the surface contact between the tool and the workpiece. Cutting forces and chip morphology during the turning process were also studied numerically by Schermann et al. [22] using ABAQUS/Explicit. It was shown that it is important to choose a fine mesh for the contact area of the tool cutting edge and the workpiece. Bagheri et al. [23] numerically and experimentally studied the effect of friction stir processing parameters and vibration on the machining behavior of the AZ91 magnesium alloy during a drilling operation. In this study, a 3D FEM modeling was applied to simulate the small-hole drilling process. The large deformation of work and the chip formation in the drilling process is realized by incorporating the Johnson–Cook material constitutive model and material failure criterion.

Moreover, with regard to the study of the dynamic behavior of thin-walled workpieces during the machining process, several approaches were investigated during the last decades. Mehdi et al. [24,25] proposed a dynamic cutting-force model for the turning process of a thin-walled workpiece in which dynamic deformations are different from those of massive ones. Due to the diversity of thin-walled workpieces in the manufacturing industry and the need for a global model of the dynamic behavior of various workpieces, the study has been oriented to characterizing the dynamic behavior of thin-walled tubular parts. The model takes into account the damping due to interference between the tool flank and the machined workpiece surface. The different tests carried out clearly show the effect of cutting damping on cutting forces and the stability of the cutting process. Lorong et al. [26], Gerasimenko et al. [27], and Luo et al. [28] have presented an experimental investigation completed with a numerical model of a straight-turning operation on a thin-walled structure. Their

works reveal the instabilities of the quasi-steady cutting under variable conditions due to the structure's mass and compliance variation. Sahraoui et al. [29] have presented an analysis of turning process stability using an analytical model supported and validated with experimental test results of roughing and finishing operations conducted on AU4G1 thin-walled tubular workpieces with different thickness values. They showed the important effects of the dynamic behavior of the thin-walled workpieces on the stability criteria, which cannot be ignored in machining process planning and cutting parameters selection. They justified the influence of additional structural damping on chatter suppression in the machining of thin-walled workpieces. Del Sol et al. [30] present a review work of thin-wall light-alloy machining, analyzing the problems related to each type of thin-wall parts, exposing the causes of both instability and deformation through analytical models, summarizing the computational techniques used, and presenting the solutions proposed by different authors from an industrial point of view. Amigo et al. [31] present an experimental investigation of the cutting forces and their prediction in high-feed turning of nickel-chrome-based superalloys. The effects of using an oil emulsion and a $CO_2$ cryogenic coolant were also studied. The results indicated a good agreement between model predictions and experimental results for three tested aerospace materials (Inconel 718, Haynes 263, and AISI 1055) using comparable cutting conditions. It was shown that while oil emulsion was the best option for Inconel 718, cryogenic cooling with $CO_2$ can open the path towards a more efficient and cleaner turning in the case of Haynes 263.

The aim of this study is to analyze the dynamic behavior of thin-wall tubes during the roughing and finishing turning process. A numerical study is proposed, supported, and validated with experimental test results to better understand the different damage observed during turning. The numerical study is based on a 3D FEM using ABAQUS/Explicit dynamics, supported by experimental tests, in order to predict the cutting forces in radial, tangential, and axial directions. In the model, the workpiece material is governed by the behavior law of Johnson–Cook.

Although the turning of 42CrMo4 steel has been experimentally studied extensively in the literature, only a few studies have dealt with the dynamic behavior of the machined thin-walled parts characterized by their low rigidity and low structural damping. It can also be stated that the numerical study of the behavior of thin-walled parts made of this material still presents an important field to be investigated. So, the novelty of this study lies in investigating the turning process stability at the machining of thin-walled tubes made of a 42CrMo4 steel alloy with the numerical approach validated experimentally. Once the numerical model is experimentally verified, it can be the basis for further studies, in which the influence of various cutting and geometrical parameters (such as cutting speed, depth of cut, feed rate, tool insert cutting angles, tubular workpiece thickness, diameter to length ratio, etc.) on the cutting forces and the stability of the machining process can be investigated without the necessity to resort to an experimental study again.

## 2. Materials and Methods

### 2.1. Numerical Process

2.1.1. Behavior Law

The turning process is accompanied by the presence of dynamic behavior not only of the cutting tool but also of the machined material, along with chip formation accompanied by high rates of deformation. For thin-walled parts, which are much less rigid, it is, therefore, very important to pay attention to the stability of the machining process, as this significantly affects tool wear and its lifetime, the quality of the machined surface, and finally, the effectiveness of the whole production process.

The simulation model within the presented research was developed on the explicit version of the ABAQUS (v. 6.3) finite element analysis software, which supports the resolution of dynamic non-linear problems with this type of condition. In this study, the workpieces are made of 42CrMo4 steel materials, which are characterized, in the case of a traction load, by an elastic behavior, a plastic behavior, and a phase of damage. During this

last phase, the mechanical properties of the material deteriorate. Analyses of experimental tests made by researchers have shown the formation of micro-cracks and micro-cavities, which lead to rupture. Therefore, the study of cutting processes requires characterizing the actual material behavior through laws describing its damage. The mechanical behavior at the deformation of the machined workpiece can be described by a thermo-viscoplastic model, which takes into account high deformation rates, inelastic deformations, and the effects linked to the variation in temperature. The behavioral law of Johnson–Cook has been implemented for the flow stress of the workpiece (Equation (1)). This law is based on the value of the equivalent plastic deformation of an element. The damage occurs when the damage parameter reaches the value 1.

$$\overline{\sigma} = \left[ A + B \left( \overline{\varepsilon}^{pl} \right)^n \right] \left[ 1 + C \ln \left( \frac{\overline{\dot{\varepsilon}}^{pl}}{\overline{\dot{\varepsilon}'_0}} \right) \right] (1 - \theta^m) \tag{1}$$

where $\overline{\sigma}$ is the equivalent plastic stress, $\overline{\varepsilon}^{pl}$ is the equivalent plastic strain, $\overline{\dot{\varepsilon}}^{pl}$ is the equivalent plastic strain rate, $\overline{\dot{\varepsilon}'_0}$ is the reference equivalent plastic strain rate, $A$ is the limit of elasticity of the material, $B$ and $n$ are the coefficients related to the work hardening, $C$ is the sensitivity coefficient to deformation rate, and $m$ is the temperature sensitivity coefficient. The constants $A$, $B$, $n$, $C$, and $m$ are given in Table 1. Their values are determined based on the flow stress data obtained from the literature according to mechanical tests [32] and $\theta^m$ is the temperature ratio given by Equation (2):

$$\theta^m = \frac{T - T_{room}}{T_m - T_{room}} \tag{2}$$

where $T$ is the workpiece temperature, $T_{room}$ is the room temperature, and $T_m$ is the melting temperature. In our study, $\theta^m$ is assumed to be zero because only the mechanical behavior of the process has been considered in the model.

**Table 1.** 42CrMo4 material parameters for the Johnson–Cook model [32].

| Parameter | Value |
|---|---|
| Young's modulus $E$ (GPa) | 211 |
| Poisson's ratio $\upsilon$ | 0.29 |
| Density $\rho$ (kg/m$^3$) | 7850 |
| $A$ (MPa) | 598 |
| $B$ (MPa) | 768 |
| $C$ | 0.0137 |
| $n$ | 0.2092 |
| $m$ | 0.807 |
| $\overline{\dot{\varepsilon}'_0}$ (s$^{-1}$) | 0.001 |

2.1.2. Friction Law Model

The interactions between the chip and the cutting tool result in friction between two surfaces. With ABAQUS, there is a "master" surface and a slave surface (geometry or mesh). The essential criterion in this contact is that the nodes relating to the slave surface do not penetrate into the "master" surface. In addition, normal vectors are calculated for each node. In the case of digital cutting simulations, the tool is defined as the master, and the workpiece is the slave.

Coulomb and Tresca friction laws are often the most used in a numerical simulation model. In the case of Coulomb's law, the two tangential and normal stresses ($\tau_t$, $\sigma_n$) exerted on the surfaces of contact are linked by the following relation (Equation (3)) in the case of a rigid tool and the deformable part:

$$\tau_t \leq \mu \, \sigma_n \tag{3}$$

where $\mu$ represents the friction coefficient.

The Tresca model imposes a constant friction threshold, and the slip limit is independent of the normal stress. In this case, the friction law is given by Equation (4):

$$\tau_t < m \frac{\sigma_e}{\sqrt{3}} \tag{4}$$

where $m$ is the Tresca coefficient ($0 \leq m \leq 1$) and $\sigma_e$ is the elastic limit of the workpiece material. In our simulation, the Coulomb model has been adopted with a friction coefficient value of 0.8 [33].

### 2.1.3. Damage Law Model

The contact zone between the cutting edge and the machined part is characterized by significant plastic deformations where the laws of the mechanics of the media contained are not applicable. For a more realistic simulation, it is necessary to use a damage model. In the damage zone (Figure 1), the stress tensor can be translated using the following relation describing the law of elasticity [20]:

$$\overline{\sigma} = (1 - \overline{D})\widetilde{\sigma} = (1 - \overline{D})\widetilde{E}(\varepsilon - \varepsilon_p) \tag{5}$$

where $\widetilde{\sigma}$ represents the tensor of effective stress, $\widetilde{E}$ represents the tensor of elasticity, $\varepsilon$ is the total strain, and $\varepsilon_p$ the viscoplastic strain. The term $\overline{D}$ represents the damage variable, which can vary according to a linear or exponential law.

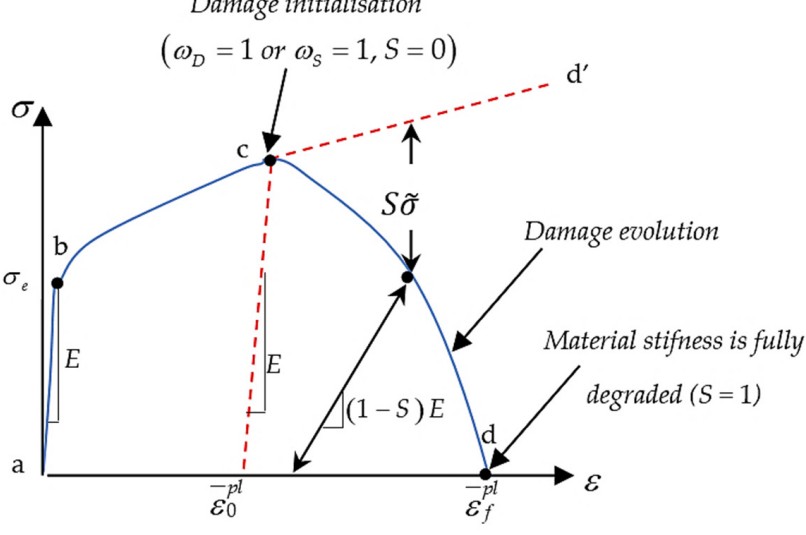

**Figure 1.** The tensile curve of a ductile metal.

The damage process is described by three phases: initiation of the damage at a deformation level and evolution of the damage accompanied by increased loading and rupture. The initiation phase of the damage $\overline{\omega}$ can be characterized by the following relationship (Equation (6)):

$$\overline{\omega} = \sum \left( \frac{\Delta \overline{\varepsilon}_p}{\overline{\varepsilon}_i} \right) \tag{6}$$

where $\Delta \overline{\varepsilon}_p$ is the increment of the equivalent plastic deformation and $\overline{\varepsilon}_i$ is the deformation equivalent to the rupture of the material. The damage is initiated if $\overline{\omega}$ takes the value 1. This term is calculated at each increment and for each mesh element.

A linear evolution of the damage $D_{lin}$ is described by the following expression (Equation (7)):

$$D_{lin} = \frac{\overline{u}}{\overline{u}_f} = \frac{\overline{u}}{2 \left( \frac{G_f}{\sigma_y} \right)} \tag{7}$$

An exponential evolution of the damage $D_{\text{exp}}$ is described by the following expression (Equation (8)):

$$D_{\text{exp}} = 1 - \exp\left(-\int_0^{\overline{u}} \frac{\overline{\sigma}}{G_f} du\right) \tag{8}$$

where $\overline{u}$ represents the equivalent plastic displacement, $\overline{u}_f$ is the fracture displacement, $\sigma_y$ is the flow stress, and $G_f$ is the rate of energy restitution of the material.

$$\left(G_f\right)_i = \left(\frac{1 - v^2}{E}\right)(K_{C,i})^2 \text{ with } i = \text{ I, II} \tag{9}$$

where $v$ is the Poisson coefficient of the material, $E$ is its Young's modulus, and $K_{C,i}$ is the factor describing the breaking strength (in mode I "crack" or II "shear") [16].

The models of damage are varied, and they essentially describe the plastic deformation equivalent to rupture. These models are generally based on constants relating to each type of material, which can be identified from experimental tests or numerical simulations. The ductile damage and the shear damage criteria were chosen within the simulation presented in the research, which describes the equivalent deformation of the material at break. The shear criterion is suitable for breaks due to the location of the shear band. The model supposes that the plastic strain equivalent at the beginning of the damage is the function of the report of shear stress and the speed of deformation. The ductile criterion of the rupture is suitable for the ruptures due to the growth and propagation of the cracks. The model supposes that the plastic strain equivalent at the beginning of the damage is a function of the three axial stresses and the rate of strain.

### 2.1.4. Dynamic Model of Cutting Forces

For a single cutting-point operation, the oblique cutting model presented in Figure 2 is used to represent the cutting force components. In this model, the tool geometry and the chip formation parameters are taken into consideration.

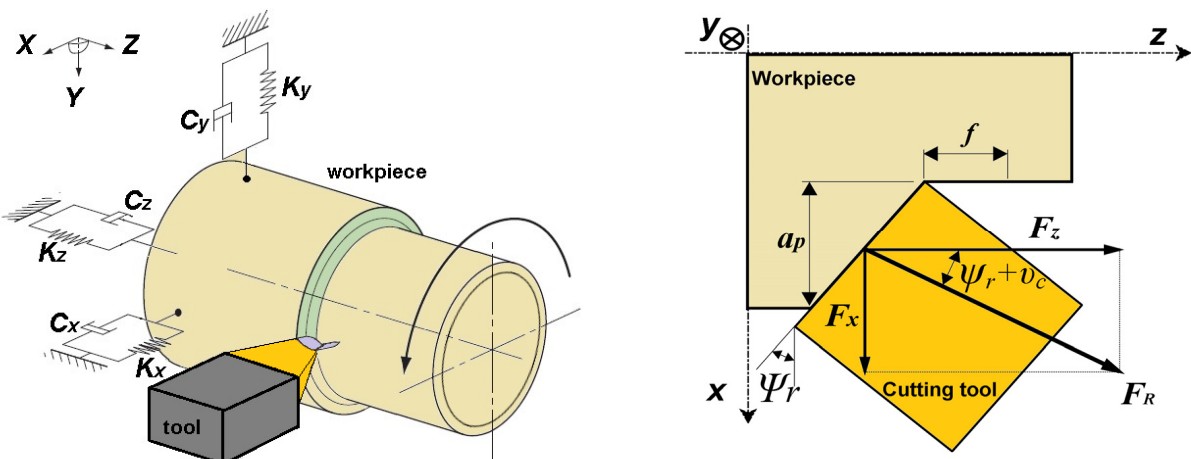

**Figure 2.** Cutting forces model description.

The resulting force acting on the structure comprises a tangential force along the cutting speed direction, a radial force along the cutting edge, and an axial force perpendicular to the cutting edge. Using geometrical parameters (chip flow angle, approach angle, . . .) these components are expressed in $x$, $y$, and $z$ coordinate system by $F_x$, $F_y$, and $F_z$ as the results of the combination between two forces: the metal-cutting force and the process-damping force.

The *first type* of force responsible for the metal-removal process is the metal-cutting force $F_c$. The three components of this force are proportional to the chip load and process vibrations. Equation (10) gives the expressions of these components:

$$
\begin{aligned}
F_{cx} &= K_R \cdot \sin(\psi_r + v_c) \cdot A_{chip} = K_{cx} \cdot A_{chip}. \\
F_{cy} &= K_T \cdot A_{chip} \qquad\qquad\;\; = K_{cy} \cdot A_{chip}. \\
F_{cz} &= K_R \cdot \cos(\psi_r + v_c) \cdot A_{chip} = K_{cz} \cdot A_{chip}.
\end{aligned}
\tag{10}
$$

where $K_R$ and $K_T$ are the cutting forces coefficients calculated according to their analytical expressions described in [24]. Note that $x$, $y$, and $z$ directions are equivalent, respectively, to the depth of cut, the cutting velocity, and the feed directions.

According to previous studies, during the cutting process, a waved surface is the result of the tool and workpiece contact mechanism caused by the regenerative effect. The cutting tool interacts with the workpiece along the clearance surface and may follow a sinusoidal path. It adds damping effects to the process, and this will also affect the vibrations. The *second type* of force is the dynamic damping force that is generated proportionally to the vibration velocity [24,34–39].

The components of the cutting damping force along $x$, $y$, and $z$ directions can be calculated according to expressions of Equation (11) [24]:

The components of the cutting damping force along $x$, $y$, and $z$ directions can be calculated according to expressions of Equation (10) [24]:

$$
\begin{aligned}
F_{dx} &= -C_R \cdot \sin(\psi_r + v_c) \cdot V_x(t) = -C_{dx} \cdot V_x(t). \\
F_{dy} &= -C_T \cdot V_y(t) \qquad\qquad\;\; = -C_{dy} \cdot V_y(t). \\
F_{dz} &= -C_R \cdot \cos(\psi_r + v_c) \cdot V_z(t) = -C_{dz} \cdot V_z(t).
\end{aligned}
\tag{11}
$$

where $V_x(t)$, $V_y(t)$, and $V_z(t)$ represent the three components of the vibration velocity of the workpiece cutting points in $x$, $y$, and $z$ directions. $C_R$ and $C_T$ represent, respectively, the repulsive and the thrust process damping coefficients. Their expressions can be calculated using Equations (12) and (13) [24]:

$$
C_R = \frac{b \cdot r \cdot B \cdot \sigma_e}{V_c \cdot \alpha_e^2}.
\tag{12}
$$

$$
C_T = \mu \cdot C_R = \mu \cdot \frac{b \cdot r \cdot B \cdot \sigma_e}{V_c \cdot \alpha_e^2}.
\tag{13}
$$

where $r$ is the cutting edge radius, $B$ is a materiel characteristic parameter, $\sigma_e$ is the yield strength of the workpiece material, $\alpha_e$ is the effective clearance angle (rad), and $\mu$ is the friction factor between the workpiece material and the tool rake face.

The effective (instantaneous) rake and clearance angle variation is opposite: when the effective rake angle $\gamma_e$ value increases, the effective clearance angle $\alpha_e$ value decreases. The angles (instantaneous) are described by the following expressions:

$$
\gamma_e = \gamma_0 + \frac{f}{\pi \cdot D}.
\tag{14}
$$

$$
\alpha_e = \alpha_0 - \frac{f}{\pi \cdot D}
\tag{15}
$$

where $\gamma_0$ is the initial rake angle, $\alpha_0$ is the initial clearance angle, and $D$ is the workpiece diameter.

### 2.1.5. Numerical Model

The finite element simulation is based on the mesh of the regions studied into elements. The distribution, the density of the mesh, and the dimensions of the elements are specified according to the problem studied. In the case of the cutting process, the mesh can never maintain its initial state. Changes are always present, and the mesh risks the problem

of distortion, leading to errors and stopping the calculation. The choice of the type of elements, the automatic correction of the mesh (adaptive mesh), and the increase in the density of the localized mesh (dimensions of the elements) are presented as techniques that help to eliminate this problem. Our simulation model is developed via the exploitation of the "Dynamic ABAQUS/Explicit" module, which makes it possible to support non-linear effects and large deformations. However, thermal effects are not taken into account due to the fact that our study mainly focuses on the cutting forces as a function of the wall thickness of the part and the variation of the feed rate.

A cantilever mounting system has been opted for in the simulation model (Figure 3). This choice is justified by the fact that the ratio of the length of the workpiece to its outer diameter is less than 2. The three jaws are modeled as a rigid body with a reference point that controls the imposed boundary conditions. The elements of mesh chosen are specific to the rigid type, and they are also of triangular type R3D3 (linear) for Explicit resolution. The density of the mesh at the surface contact respects that adopted for the part. The boundary conditions are imposed on the tool and the three jaws through their reference points. The tool has a constant feed rate along the z-axis. The jaws turn at a constant angular speed around the feed rate axis. The rotational movement is transmitted to the workpiece through the jaws' surface contact. For the contact properties, the part is considered the "slave", and the jaws and the cutting tool are considered the "master". The type of contact chosen is "General contact" with two types of properties: The tool and workpiece contact is with friction translated by the introduction of a friction coefficient $\mu$. The jaws and workpiece contact are non-slip to transmit the rotational movement of the jaws toward the workpiece. The material properties taken into account in the simulation are shown in Table 1. The cutting parameters for marching 42CrMo4 steel parts considered in the numerical simulation and experimental tests are shown in Table 2.

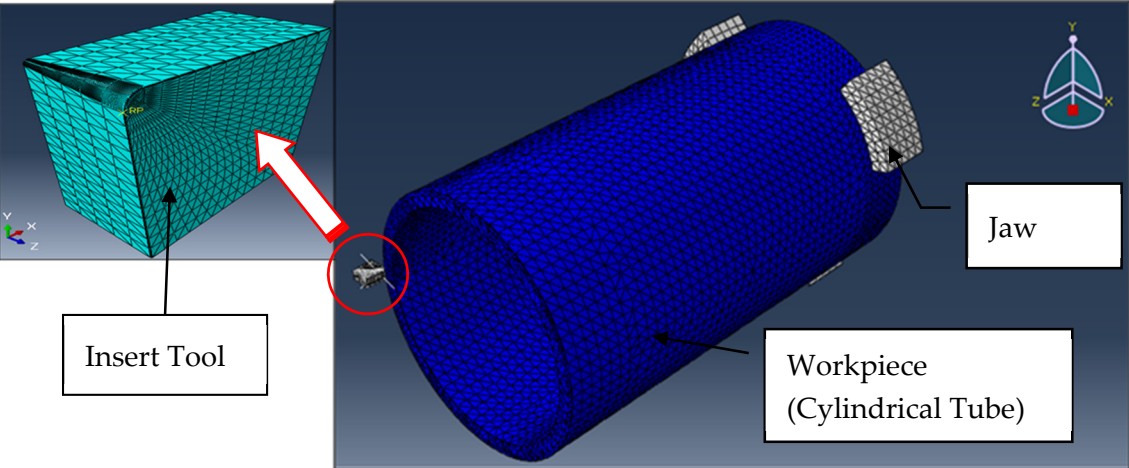

**Figure 3.** Model of the cutting system.

**Table 2.** Considered input parameters in numerical and experimental tests for 42CrMo4 steel alloy.

| | Machining Operation | | | | | | | | | | | |
|---|---|---|---|---|---|---|---|---|---|---|---|---|
| | Roughing | | | | | | | | | Finishing | | |
| Test Ref. | R-1 | R-2 | R-3 | R-4 | R-5 | R-6 | R-7 | R-8 | R-9 | F-1 | F-2 | F-3 |
| Wall thickness $e_p$ (mm) | 6 | 6 | 6 | 4.5 | 4.5 | 4.5 | 3 | 3 | 3 | 1.5 | 1.5 | 1.5 |
| Depth of cut $a_p$ (mm) | 1.5 | 1.5 | 1.5 | 1.5 | 1.5 | 1.5 | 1.5 | 1.5 | 1.5 | 0.5 | 0.5 | 0.5 |
| Feed rate $f$ (mm/rev) | 0.2 | 0.3 | 0.4 | 0.2 | 0.3 | 0.4 | 0.2 | 0.3 | 0.4 | 0.1 | 0.125 | 0.16 |
| Cutting speed $v_c$ (m/min) | 160 | 160 | 160 | 160 | 160 | 160 | 160 | 160 | 160 | 200 | 200 | 200 |

The tool is considered a non-deformable (rigid) body. The chosen elements are specific to the rigid type (discrete rigid element) and are of quadratic type R3D4 (linear) for explicit

resolution. The density of the mesh is important at the level of the cutting spout of the tool in order to respect the tool and workpiece contact (Figure 3). Numerical simulation concerns only the steel alloy 42CrMo4 governed by the law of Johnson–Cook; the details of the geometry of the tool are presented in Table 3. Tool references are CNMG- 120408-QMGC-415 and CNMG-120404-QFGC-425, respectively, for roughing and finishing operations.

**Table 3.** Geometric characteristics of cutting tools (roughing and finishing of 42CrMo4 steel parts).

| Parameters | Machining Operation | |
|---|---|---|
| | **Roughing** | **Finishing** |
| Tool insert reference (ISO) | CNMG 120408 QMGC 415 | CNMG 120404 QFGC 425 |
| Tools noise radius: $R$ (mm) | 0.8 | 0.4 |
| Acuity radius of the cutting edge: $r$ (mm) | 0.04 | 0.03 |
| Rake angle: $\gamma$ (°) | 7 | 2 |
| Clearance angle: $\alpha$ (°) | 6 | 6 |
| Cutting edge angle: $\chi_r$ (°) | 95 | 95 |
| Complementary angle of the cutting edge direction: $\psi_r$ (°) | −5 | −5 |

The workpieces are modeled with thin tubes with a length "$L$" equal to 200 mm and an internal diameter "$D$" equal to 100 mm. The wall thicknesses "$e_p$" depends on the machining operation: For a roughing operation, the considered thicknesses are 3, 4.5, and 6 mm. And for a finishing operation, the wall thickness is 1.5 mm. The workpieces are considered to be deformable and meshed with linear tetrahedral elements C3D4. The highest mesh density is located in the zone contact between the workpiece and the cutting tool zone, where the mesh size is between 0.1 mm and 0.01 mm. In the remaining zones, a coarse mesh is applied with a size of around 5 mm in order to reduce the element number and, therefore, time computation. The default ALE-adaptive mesh parameters were adopted in this simulation. The smoothing algorithm is determined using an analysis product with a meshing predictor set to the current deformed position because it is recommended for problems with very large distortions. The advection parameter is set to second order to use a second-order algorithm to remap solution variables after adaptive meshing has been implemented. The time step adopted in the simulation is 0.5 ms. The time step is small enough to collect the data in different valuable steps.

### 2.2. Experimental Procedure

The experimental tests were conducted with roughing and finishing operations using tubular parts with a 200 mm length and an internal diameter of 100 mm. As a workpiece material, the steel alloy 42CrMo4, which is used in mechanics for parts of different sizes (shafts, racks, crankshafts, gears, etc.) due to its good mechanical characteristics, was selected In total, twelve experimental tests were performed: nine tests for the roughing operation and three tests for the finishing operation. Only one test was carried out per piece and per tool insert in order to minimize the effect of tool wear on the variation of cutting forces. In fact, wear is a consequence of the interaction between the cutting tool, the material of the workpiece, and the cutting conditions of machining, in which the wear mechanism is characterized by the abrasion of elementary particles of the interface contact layers and their removal in the form of wear products from the cutting zone (most often in the form of chips). Overall wear of the cutting edge is usually the result of abrasion, plastic deformation, and brittle failure. This is, therefore, an unwanted phenomenon, the manifestation of which can be secondary vibrations, and this was not the aim of the research (just as the research was not focused on the origin and formation of chips, as this will be part of the research carried out later). Therefore, a new cutting tip was always used for individual machining operations, and in such a way that signs of wear did not have time to affect the measurements.

The experiments consisted in measuring firstly the cutting force components $F_x$, $F_y$, and $F_z$ along radial, tangential, and axial directions, respectively, and secondly, the tangential workpiece vibrations along the $y$ direction. The experimental setup is illustrated in Figure 4. The measurement of the cutting forces components along the three directions $x$, $y$, and $z$, is conducted using a three-axial force piezoelectric dynamometer, which can measure the cutting force components signals in three directions during the active cutting process. The workpiece vibration along the y direction is measured using a capacitive sensor without contact fixed at a distance of 0.5 mm to the free end of the workpiece. The engine lathe is equipped with a 20 kW DC motor. The rotation speed is controlled using an electronic regulator. The signals received from the dynamometer and the sensor are acquired using a data acquisition system and analyzed using Matlab software (V 7.0.1.24704 R14).

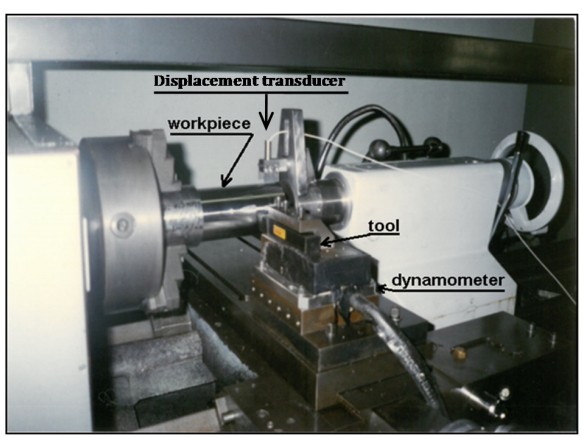

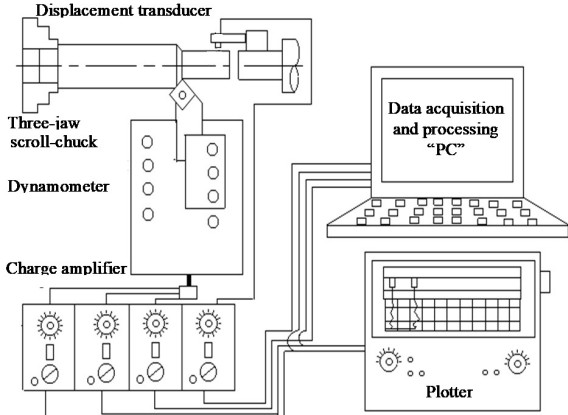

**(a) Detail of the turning machine**     **(b) Data acquisition and processing chain**

**Figure 4.** Experimental set.

## 3. Results and Discussions

### 3.1. Analysis of the Cutting Force and Radial Vibration of the Workpiece Wall

The experimental results of the twelve performed tests (with the parameters described in Table 2) are summarized in Table 4. The observation and analysis of the different traces marked by the cutting tool on the machined part surface of the workpiece (Figure 5) show that in the case of the vibrated process, two types of weary surfaces and two zones can be observed on the lateral surface. The first type of weary surface (Type 1) is complex. It presents multiple shock effects on the surface. This first type corresponds to a heavy chattering phenomenon for which the cutting tool does not work in continuous conditions, and the contact appears to loosen intermittently. This surface is located in a first zone (zone 1) from 0 to 60 mm along the revolution $z$-axis. This zone constitutes the most distant from the chuck.

**Table 4.** Average values of experimental cutting forces and radial displacements of workpiece wall.

| Test Ref. | $f$ (mm) | $F_x$ (N) | $F_y$ (N) | $F_z$ (N) | $dep$ (µm) | Observed Surface |
|---|---|---|---|---|---|---|
| R-1 | 0.2 | 260 | 791 | 391 | −61 | N. V. |
| R-2 | 0.3 | 360 | 1091 | 491 | −95 | N. V. |
| R-3 | 0.4 | 482 | 1465 | 574 | −132 | N. V. |
| R-4 | 0.2 | 255 | 780 | 392 | −63 | N. V. |
| R-5 | 0.3 | 319 | 1040 | 455 | −81 | N. V. |
| R-6 | 0.4 | 530 | 1408 | 684 | −119 | N. V. |
| R-7 | 0.2 | 256 | 790 | 387 | −69 | N. V. |
| R-8 | 0.3 | 359 | 1076 | 505 | −119 | N. V |
| R-9 | 0.4 | 492 | 1462 | 662 | −135 | L. V. |
| F-1 | 0.1 | 91 | 249 | 141 | −39 | V. |
| F-2 | 0.125 | 98 | 250 | 100 | −34 | V. |
| F-3 | 0.16 | 156 | 307 | 168 | −49 | V. |

N. V. = no vibrations; L. V. = light vibrations; V. = vibrations.

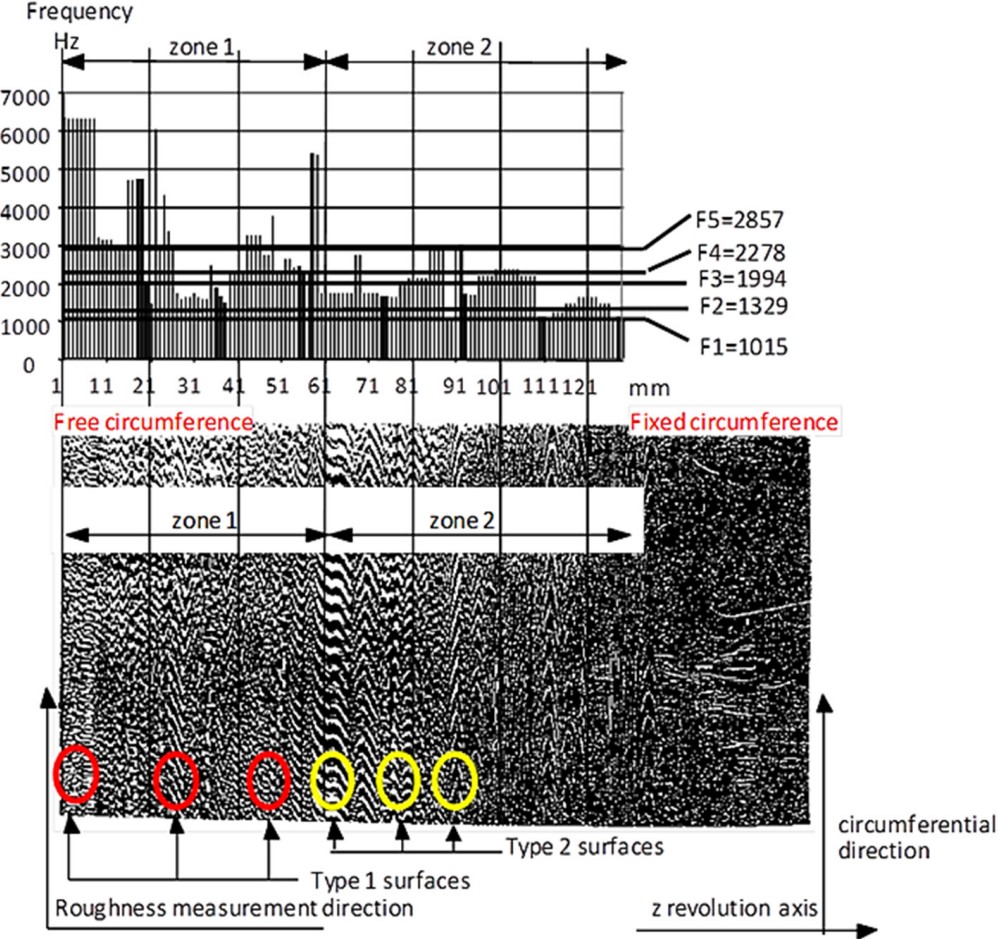

**Figure 5.** Analysis of the machined outer surface of the tube (finishing, wall thickness $e_p$ = 1.5 mm).

The second type (Type 2) corresponds to regular waves, for which the cutting tool is considered to work with a varying chip cross-section but with continuous contact. This type of surface is located in a second zone (zone 2), which is close to the chuck.

Table 5 summarizes the average values of the radial, tangential, and axial cutting forces $F_x$, $F_y$, and $F_z$ obtained experimentally and numerically based on the feed rate during the roughing and finishing operations. The experimental values of $F_x$, $F_y$, and $F_z$ are denoted, respectively, $F_{x\_exp}$, $F_{y\_exp}$, and $F_{z\_exp}$. Those obtained using the analytical model, are denoted, respectively, $F_{x\_sim}$, $F_{y\_sim}$, and $F_{z\_sim}$. Finally, those obtained using the FEM are denoted, respectively, $F_{x\_fem}$, $F_{y\_fem}$, and $F_{z\_fem}$. The values given in brackets indicate the percentage error calculated between the cutting forces obtained numerically and those measured experimentally. In the case of roughing operations, the values of the experimental tests and those obtained using the analytical model show agreement in the results of the three components of the total cutting force. The recorded values of errors do not exceed 18% in the case of the axial component $F_z$ relating to the roughing test R-5 and 26% in the case of the radial component $F_x$ relating to the roughing test R-6. During finishing operations, it was noted that there was an agreement between the cutting forces measured experimentally and those simulated analytically. In the case of test F-2, a difference of 11% was recorded between the axial force measured experimentally and that simulated analytically. However, it can be stated that the average values of the radial, tangential, and axial cutting forces simulated using the FEM are lower than those measured experimentally. These can be explained using the results discussed by Bargue [40] and Arrazola et al. [41]. Thus, according to Barge [40], the size of the mesh elements has an influence on the cutting forces and the radius of curvature of the chips, where extreme refinement can cause localization problems. Depending on the size of the mesh, the simulation results can sweep

different morphologies of chips obtained experimentally. A decrease in the mesh size causes an uncontrollable increase in the values of the thrust-cutting force and the cutting energy [41].

**Table 5.** Average values of experimental and numerical cutting forces *Fx, Fy*, and *Fz* during roughing and finishing operations.

| | $F_{x\_exp}$ | $F_{x\_sim}$ (err%) | $F_{x\_fem}$ (err%) | $F_{y\_exp}$ | $F_{y\_sim}$ (err%) | $F_{y\_fem}$ (err%) | $F_{z\_exp}$ | $F_{z\_sim}$ (err%) | $F_{z\_fem}$ (err%) |
|---|---|---|---|---|---|---|---|---|---|
| R-1 | 260 | 277 (6) | 53 (80) | 791 | 787 (1) | 774 (2) | 391 | 381 (3) | 239 (39) |
| R-2 | 360 | 359 (0) | 63 (82) | 1091 | 1088 (0) | 1034 (5) | 491 | 495 (1) | 263 (47) |
| R-3 | 482 | 435 (10) | 94 (81) | 1465 | 1445 (1) | 1271 (13) | 574 | 599 (4) | 312 (46) |
| R-4 | 255 | 211 (17) | 23 (91) | 780 | 779 (0) | 800 (3) | 392 | 406 (4) | 244 (38) |
| R-5 | 319 | 280 (12) | 46 (86) | 1040 | 1044 (0) | 900 (13) | 455 | 537 (18) | 260 (43) |
| R-6 | 530 | 390 (26) | 77 (85) | 1408 | 1393 (1) | 1225 (13) | 684 | 748 (9) | 304 (55) |
| R-7 | 256 | 217 (15) | 30 (88) | 790 | 790 (0) | 705 (11) | 387 | 416 (8) | 231 (40) |
| R-8 | 359 | 292 (19) | 35 (90) | 1076 | 1079 (0) | 1005 (7) | 505 | 559 (11) | 241 (52) |
| R-9 | 492 | 406 (18) | 96 (81) | 1462 | 1416 (3) | 1350 (8) | 662 | 777 (17) | 310 (53) |
| F-1 | 91 | 89 (2) | 17 (81) | 249 | 248 (0) | 176 (29) | 141 | 148 (5) | 93 (34) |
| F-2 | 98 | 96 (3) | 22 (78) | 250 | 246 (2) | 190 (24) | 100 | 110 (11) | 93 (7) |
| F-3 | 156 | 157 (0) | 23 (85) | 307 | 306 (1) | 216 (30) | 168 | 181 (8) | 99 (41) |

The experimental results show that:

1. The average values of the cutting forces and the radial displacement of the workpiece wall are proportional to the chip cross-section $a_p \times f$ (mm$^2$). These average values are slightly influenced by the wall thickness of the machined parts. For the finishing tests (small chip cross-section), the average values are lower than those for the roughing tests (large chip cross-section). Indeed, an increase in the feed rate $f$ from 0.2 mm/rev to 0.4 mm/rev is accompanied by a significant increase in the average values of $F_x$, $F_y$, and $F_z$: for the workpiece of 3 mm thick, $F_x$, $F_y$, and $F_z$ increase by 102%, 80%, and 79%, respectively. For the workpiece of 4.5 mm thick, the increase in $F_x$, $F_y$, and $F_z$ is, respectively, 104%, 78%, and 75%. Finally, for the workpiece of 6 mm thick, $F_x$, $F_y$, and $F_z$ increased by 102%, 79%, and 78%, respectively;

2. The variation of the radial, tangential, and axial components $F_x$, $F_y$, and $F_z$ of the cutting force, as well as the radial displacement of the workpiece wall around their average value, is low for the roughing tests where the wall thickness $e_p \geq 3.5$ mm. However, this variation intensifies for the finishing tests where the wall thickness $e_p = 1.5$ mm. These variations have a significant effect on the quality of the surface finish of the workpiece wall;

3. For the roughing operations, no vibrated surfaces were observed except for test "R-9". In this case, the flexibility of the workpiece limits the chip cross-section value without vibrations. For the other tests, variations of the dynamic cutting forces were measured. The variations are situated from ±10% to ±23% of the average value.

From these results, it can be concluded that the cutting forces and the quality of the machined surface depend not only on the choice of cutting parameters but also on the dynamic behavior of thin-walled parts due to their low rigidity and low structural damping during the machining operation. Figure 6 shows the variation of the tangential cutting force and of the radial displacement of the wall of the part, respectively, for the roughing test "R-3" (Figure 6a) and for the finishing test "F-2" (Figure 6b). In roughing, the cutting process is accompanied by low amplitude vibrations, while in finishing, the vibrations are intense.

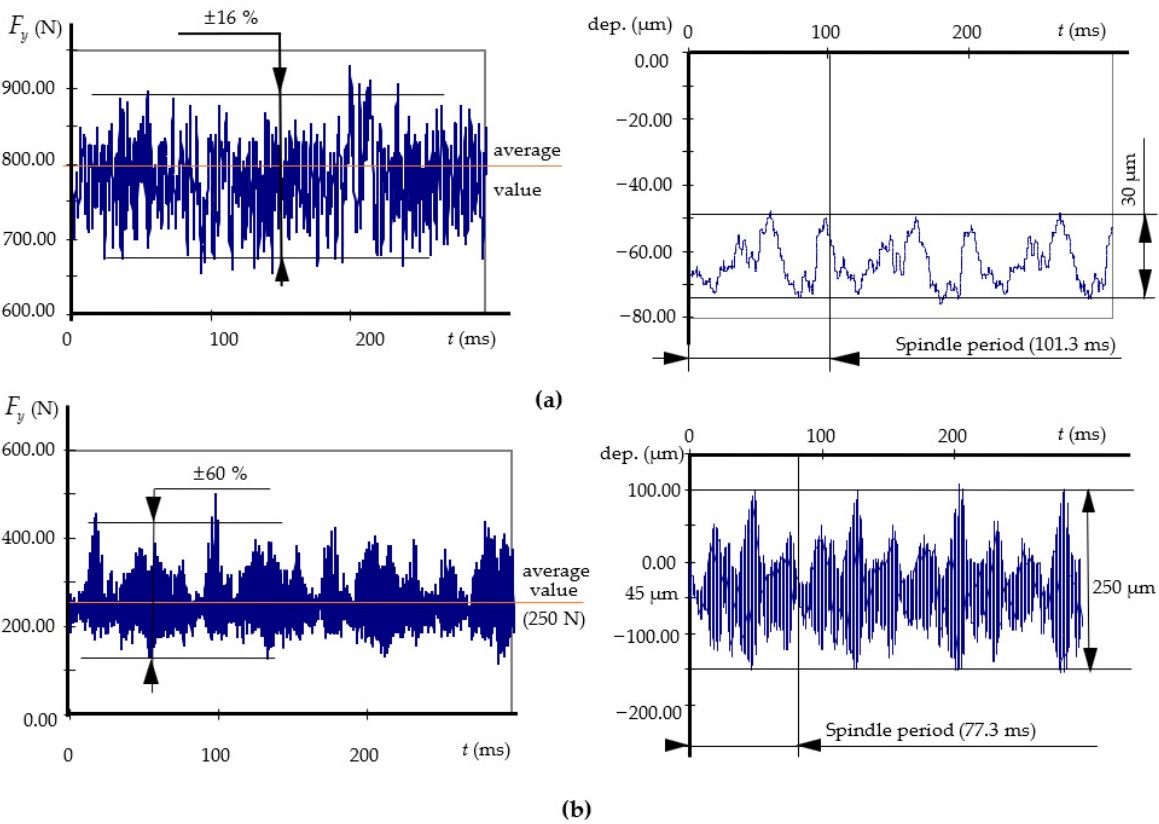

**Figure 6.** Tangential cutting force and radial displacement of the workpiece wall during machining: (**a**) roughing test "R-4"; (**b**) finishing test "F-2".

*3.2. Stability Analysis*

Chatter is a harmful phenomenon in the cutting process, which can cause bad machining quality for the work part and accelerate the wear and damage of the cutting tool. In essence, the chatter is attributed to the self-excited vibration in the machining system. The chatter could be stimulated due to the regeneration effect of chip thickness or mode coupling in different directions, and so on. However, the regenerative chatter appears earlier than other chatters in most cutting processes [42]. Therefore, the regenerative effect is considered to be the main type of chatter in metal cutting, which is due to phase differences between the vibration waves left on both sides of the chip [43].

The stability zones within this research were analyzed using the stability lobe diagrams. These diagrams are based on a theoretical model capable of determining the stability zones of the cutting process in order to choose the appropriate cutting depth of a cut for a given spindle frequency. This cutting depth is located between a minimum theoretical value (zero) and a critical limit value. Beyond this limit value, the process becomes unstable. Figure 7 presents stability lobe diagrams relative to the considered input parameters in numerical and experimental tests for the 42CrMo4 steel alloy (Table 2). These diagrams were constructed using the equation of the critical depth of cut at a stability boundary limit and the spindle speed (*N*): (Equation (16) [29])

$$a_{p\lim} = \frac{-1}{2 \cdot \left[ G_{mRx} \cdot K_{Rx} \cdot \tan(\psi_r) + G_{mRz} \cdot K_{Rz} \right]}. \tag{16}$$

where $a_{p\lim}$ is the critical depth of cut at the stability boundary limit, $\psi_r$ is the tool approach angle, $K_{Rx}$ and $K_{Rz}$ are the repulsive cutting forces coefficients in $x$ and $z$ directions, and $G_{mRx}$ and $G_{mRx}$ are the repulsive normalized dynamic compliance of the cutting structure in $x$ and $z$ directions. The reader can find more details on the solution methodology used to generate the stability lobes model in our previous work [29].

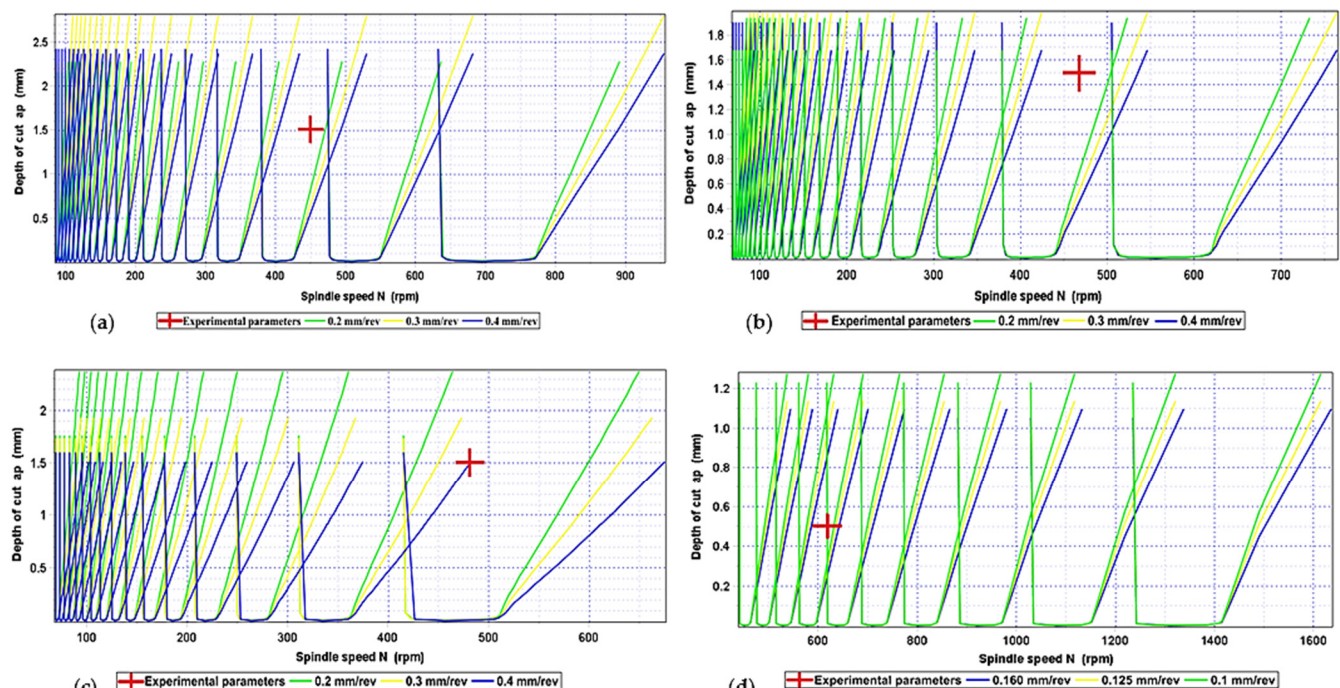

**Figure 7.** Comparison of the vibratory state of the roughing and finishing tests with the stability lobes: (**a**) tests R-1, R2, and R3, (**b**) tests R-4, R-5, and R-6, (**c**) tests R-7, R-8, and R-9, and (**d**) tests F-1, F-2, and F-3.

The stable zone is located below the curve, while the unstable zone is above (Figure 7). For neighboring curves, we may have two different zones of intersection, stable and unstable, in the same plan. The total stability boundary is defined as the continuity of the successive branches of the adjacent curves limited by their intersection points.

From Figure 7 and for a workpiece 6 mm thick, corresponding to roughing tests R-1, R-2, and R-3 (Figure 7a), it can be noted that for a depth of cut of 1.5 mm and for the two feed rates of 0.2 and 0.3 mm/rev, the cutting process is stable for a spindle speed of 625 rpm. For the three values of the feed rate 0.2, 0.3, and 0.4 mm/rev, a zone of stability is located for a depth of cut equal to 1.3 mm and a spindle speed equal to 630 rpm. It is noted that for the roughing experimental tests, the values retained for the spindle speed (450.7 rpm) and the depth of cut (1.5 mm) are located in unstable zones.

In the case of the roughing operation of the workpiece, 4.5 mm thick (Figure 7b) corresponding to the three roughing tests R-4, R-5, and R-6, the cutting depth allowing a stable zone is limited to 1.4 mm with a spindle speed of 500 rpm and a feed rate equal to 0.2 mm/rev. The choice of a cutting depth equal to 1 mm is possible for the two feed rates of 0.2 and 0.3 mm/rev and a spindle speed of 500 rpm. For the three values of the feed rate 0.2, 0.3, and 0.4 mm/rev, a depth of cut equal to 0.8 mm is located in the stable zone with a spindle speed equal to 500 rpm.

In the case of the roughing operation of the workpiece 3 mm thick (Figure 7c) corresponding to the three roughing tests, R-7, R-8, and R-9, in order to migrate to the stable zones, it is necessary to increase the spindle speed and reduce the depth of cut. To maintain low spindle speeds, the depth of cut should be less than 1 mm. This value is reached for the feed 0.2 mm/rev and a spindle speed between 410 and 415 rpm.

In the case of the finishing operation of the workpiece 1.5 mm thick (Figure 7d) corresponding to the three finishing tests F-1, F-2, and F-3, in order to migrate to stable zones, it is necessary to increase the spindle speed and decrease the depth of cut. For a depth of cut equal to 0.5 mm, the stable zones are located for a spindle speed greater than 1000 rpm.

### 3.3. FFT Spectra Analysis

Figure 8 represents the FFT spectra of the experimental radial vibrations ($U_y$) of the workpiece wall during roughing and finishing operations. In the FFT spectrum, there is a difference in scales between the different tests according to their vibratory state discussed in Section 3.1 and their stability (or instability) degree discussed in Section 3.2. In fact, in the roughing cases (less unstable process), the amplitudes are much lower at the peak of the dominant frequencies than those obtained in the finishing case (more unstable process). These amplitudes are considered parameters to categorize stable and unstable processes.

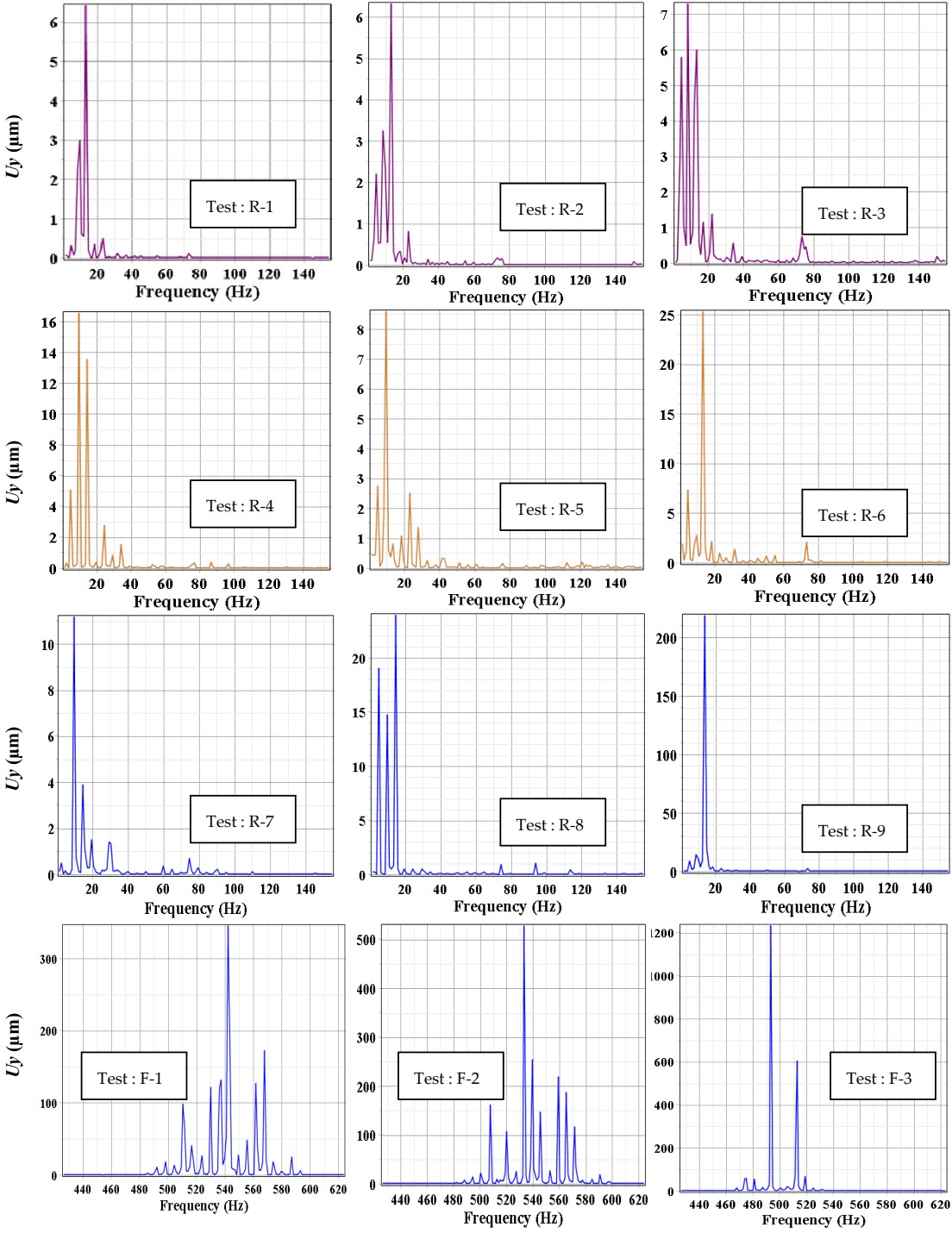

**Figure 8.** FFT spectra of radial vibrations ($U_y$) of the workpiece wall during experimental roughing and finishing operations.

In summary, for these two operations, the vibrations of the workpiece wall in the tangential direction at the free end are proportional to the feed rate. In roughing, the wall of the machined part vibrates at low frequencies, which do not exceed 15 Hz in the vicinity of the spindle rotation frequency. While finishing, the vibration frequencies are high in the vicinity of 550 Hz, which is considered high compared to the spindle frequency of 10.3 Hz (618 rpm). In fact, in the finishing case, chatter occurred due to workpiece flexibility, and the recorded spectra are made up of several harmonics that occurred at around 550 Hz. In addition, the amplitude of the peaks, which describes the level of power contained in the signals, shows an increase similar to that of the amplitudes of the temporal signal.

Indeed, the experimental results indicate that an increase in the feed rate is accompanied by a considerable amplification of the vibrations of the machined workpiece wall. Thus, the response of the cutting system depends on the choice of feed rate and the thickness of the workpiece wall. This result shows the importance of the choice of cutting parameters (low feed rate) and the rigidity of the workpiece on the stability of the cutting process.

To identify the resonant frequencies and potential instability of the tube itself, modal analyses were performed in the ANSYS 2023R2 software (Figure 9), while the following data were used for the material characterization [44]: $Rp_{0.2}$ = 750 MPa; Rm = 1100 MPa; Poisson's ratio 0.3; and density $\rho$ = 7.72 kg/dm$^3$. One end of the tube was fully constrained to simulate clamping.

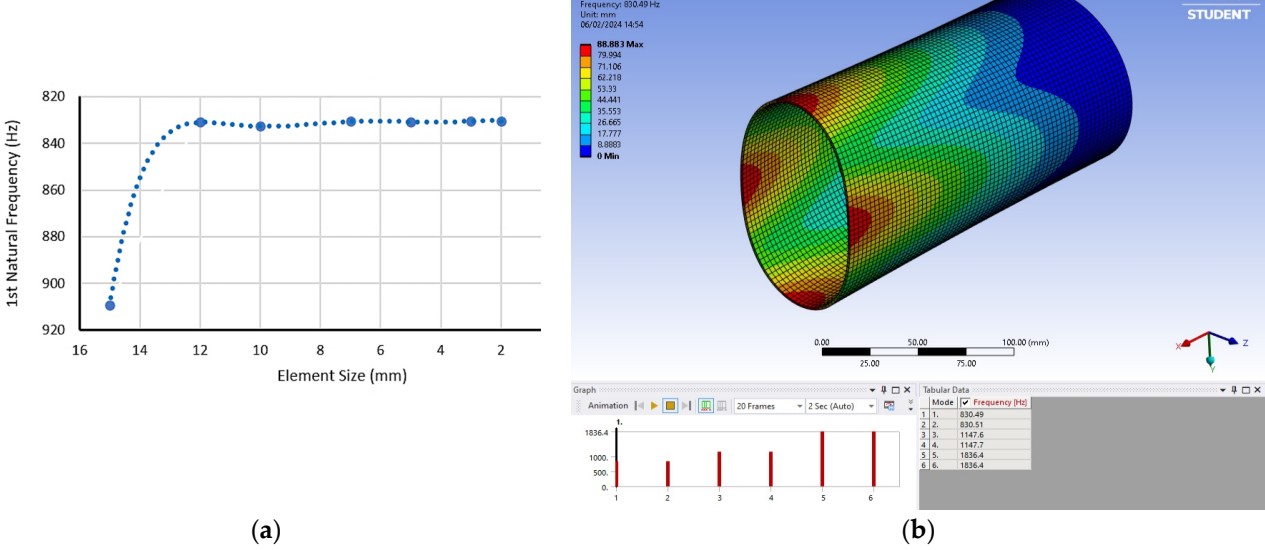

(**a**)                                          (**b**)

**Figure 9.** Modal analysis of the workpiece 1.5 mm thick; (**a**) sensitive analysis of the element size; and (**b**) first natural frequency and first mode shape results.

The lowest natural frequency, $f_1$ = 830.49 Hz, was identified for the least stiff workpiece where the considered wall thickness was 1.5 mm; therefore, the analysis results of this workpiece were selected for presentation in this manuscript. A cubic element of size 5 mm was used for the numerical solution because the sensitive analysis (Figure 9a) showed a convergent behavior of the dependence of frequency—element size, and this element size was sufficiently accurate for the solution. Within the network, 7303 elements with 50,854 nodes were created for analysis.

The results of the modal analysis showed that the natural frequencies are much higher than those achieved experimentally, which confirmed that the instability of the cutting process under the given conditions is not caused by the resonance of the workpiece but is influenced by the parameters of the technological process.

### 3.4. Von Mises Stress Field Distribution

Figure 10 describes the von Mises stress field distribution relating to the numerical roughing and finishing tests. From this figure, the most important stress values are recorded at the cutting-edge contact of the tool and workpiece. This causes the progressive degradation of the properties of the material (rigidity) and, therefore, the elimination of the elements. The other areas of the part (which will form the future chip) are also subjected to significant stresses, but they reach the maximum value only in the event of contact with the tool. However, the authors' interest has been focused on the prediction of cutting forces. Thus, a reduction in the value of the maximum degradation parameter of the mesh elements was chosen. As a result, the chip morphology was not observed in the simulation. According to Figure 10, primary shear zones (at the level of the shear plane) where the material undergoes permanent and secondary plastic deformation (at the level of the cutting surface) between the tool and the chip were distinguished. Their distributions around the contact zone are variable due to variations in the contact surfaces in movements between the tool and the part.

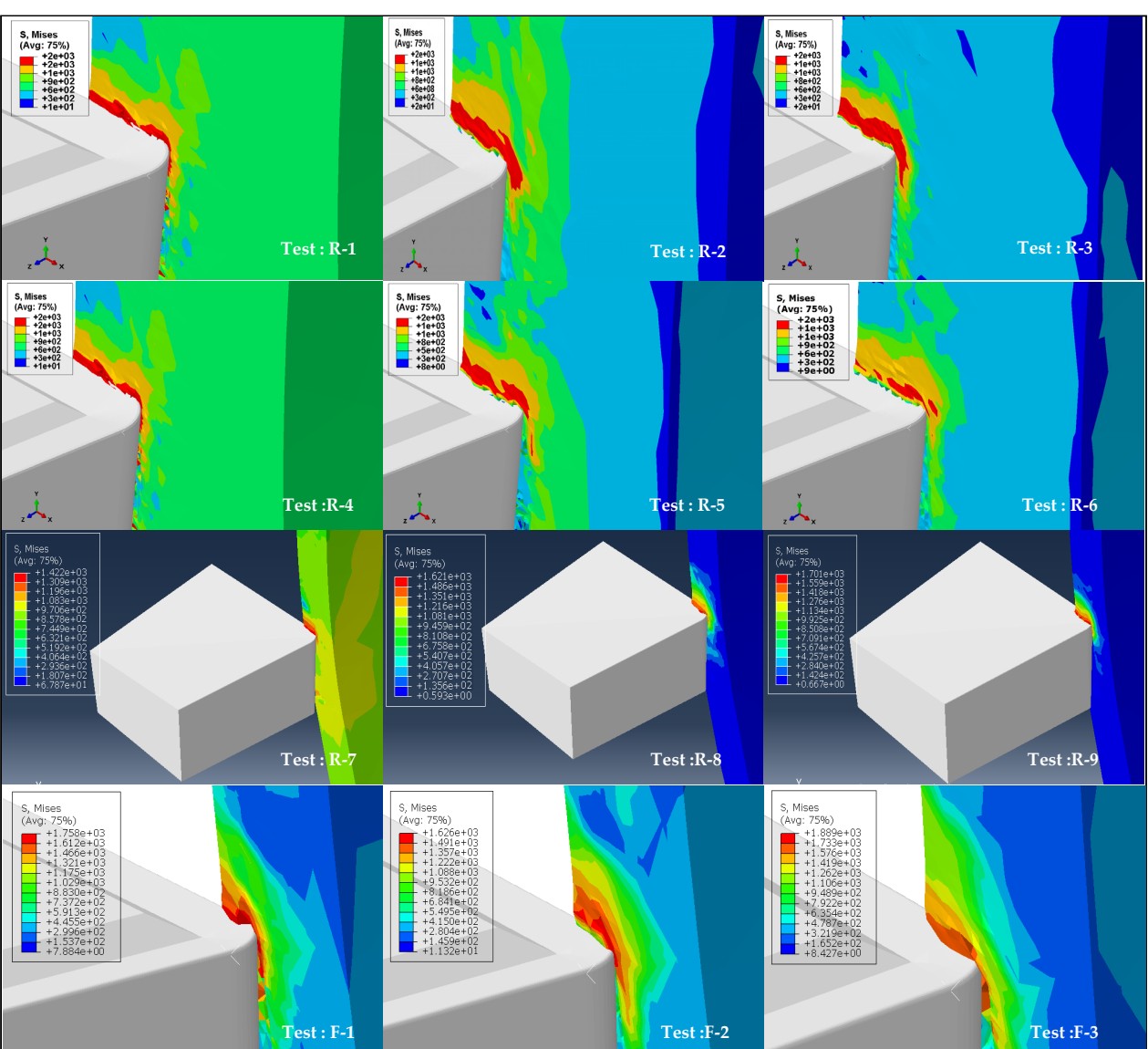

**Figure 10.** Simulated von Mises equivalent stresses (MPa) for roughing and finishing operations.

From Figure 10, it can be seen that, in the roughing operation of parts that are 3 mm thick, the stress increases when the feed rate increases. And for workpieces of 4.5 and 6 mm

thickness, the stress is not too much influenced by the feed rate. For the finishing operation, the stresses reach their maximum values at the cutting-edge contact of the tool and the workpiece. Also, the primary and secondary shear zones can be distinguished.

## 4. Conclusions

This work focuses on the dynamic behavior of thin-walled workpieces made of a 42CrMo4 steel alloy during the turning process. The impact of feed rate and the thickness of the wall part on the cutting forces and workpiece displacements has been investigated numerically and experimentally during roughing and finishing operations. The simulated cutting forces and the wall-workpiece displacements are validated according to a series of experiments performed in the same cutting conditions. In the FEM, the cutting tool is considered to be a rigid body (to simplify the model and reduce calculation time) and animated with a forward movement at a constant speed. However, the workpiece, animated with rotational movement, was considered a deformable body to simulate its rotation and dynamic behavior.

From this investigation, the following conclusions can be drawn.

1.  Cutting forces and the quality of the machined surface depend not only on the choice of the feed rate but also on the dynamic behavior of thin-walled parts due to their low rigidity and low structural damping during the machining operation;
2.  The average values of the cutting forces and the radial displacement of the workpiece wall are proportional to the feed rate. These average values are slightly influenced by the wall thickness of the machined parts;
3.  The results of the modal analysis confirmed that the instability of the cutting process under the given conditions is not caused by the resonance of the workpiece but is influenced by the parameters of the technological process;
4.  The variation of the radial, tangential, and axial components of the cutting force, as well as the radial displacement of the workpiece wall around their average value, is low for the roughing tests where the wall thickness is considered large. However, this variation intensifies for the finishing tests where the wall thickness is thin. These variations have a significant effect on the quality of the surface finish of the workpiece wall;
5.  The stress distribution is concentrated in the cutting zone. It reaches its maximum value at the cutting-edge contact of the tool and workpiece. Analysis of results allowed deducing the existence of shear zones at the edge of the tool throughout the cut.

All these results show the important effects of the dynamic behavior of the thin-walled workpieces on the stability criteria, which cannot be ignored in machining process planning and cutting parameters selection. Our main future work concerns improving the numerical model results by integrating the effect of temperature, tool wear, and the variation of the specific cutting energy during the turning operation.

**Author Contributions:** Conceptualization, K.M. (Kamel Mehdi), Z.S. and K.M. (Katarina Monkova); methodology, K.M. (Kamel Mehdi) and K.M. (Katarina Monkova); software, K.M. (Kamel Mehdi), Z.S., N.G., J.K. and K.M. (Katarina Monkova); validation, K.M. (Kamel Mehdi), P.P.M. and K.M. (Katarina Monkova); formal analysis, K.M. (Kamel Mehdi), K.M. (Katarina Monkova) and J.K.; investigation, K.M. (Kamel Mehdi), Z.S., N.G. and K.M. (Katarina Monkova); resources, K.M. (Kamel Mehdi) and Z.S.; data curation, K.M. (Kamel Mehdi); writing—original draft preparation, K.M. (Kamel Mehdi), Z.S., N.G. and K.M. (Katarina Monkova); writing—review and editing, K.M. (Kamel Mehdi), K.M. (Katarina Monkova) and P.P.M.; visualization, K.M. (Kamel Mehdi), Z.S., N.G. and K.M. (Katarina Monkova); supervision, K.M. (Kamel Mehdi), P.P.M. and K.M. (Katarina Monkova); project administration, K.M. (Kamel Mehdi). and P.P.M.; project acquisition, K.M. (Kamel Mehdi) and P.P.M. All authors have read and agreed to the published version of the manuscript.

**Funding:** The article was prepared thanks to the support of the Ministry of Higher Education and Scientific Research of the Tunisia Republic and the Ministry of Education, Science, Research, and Sport of the Slovak Republic through the grant KEGA 032TUKE-4/2022.

**Data Availability Statement:** Data are contained within the article.

**Acknowledgments:** The article was prepared thanks to the support of the Ministry of Higher Education and Scientific Research of the Tunisia Republic and the Ministry of Education, Science, Research, and Sport of the Slovak Republic through the grant APVV-19-0550.

**Conflicts of Interest:** The authors declare no conflicts of interest.

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
