# Peer review of "Investigation of Dynamic Behavior and Process Stability at Turning of Thin-Walled Tubular Workpieces Made of 42CrMo4 Steel Alloy"

_machines, doi:10.3390/machines12020120_

Round 1

Reviewer 1 Report

Comments and Suggestions for Authors

A brief summary:

The aim of the paper was to investigate the dynamic behavior and stability of the turning process of thin-walled tubular parts made from 42CrMo4 steel. The authors conducted the numerical analysis based on three-dimensional finite element method as well the experimental tests. The main aim of the paper was to investigate the influence of cutting parameters on cutting force components and stability of the process.

Broad comments:

A significant achievement of the study is the investigation of the dynamic behavior of the thin-walled workpieces on the stability criteria of the process, which is important in terms of proper planning and selection of technological parameters. The subject of the paper is important as it can expand the knowledge of the behavior of thin-walled workpieces under various machining conditions. The paper structure is fine, the methods and results are described in a clear way. The references are numerous and mostly up-to-date. The editing of the paper is fine, although there are some minor errors and typos that should be addressed.

Specific comments:

·        As authors investigated the stability of the turning of thin-walled parts, one of the best and most important indicators of the stability of the process is the surface quality of the workpiece. Why authors decide not to measure any surface roughness parameter after turning? In some instances, authors refer to the surface quality of the workpieces but there is no data on any measurement nor any mention on how it was assessed (i.e. line 393 “…have a significant effect on the quality of the surface finish of the workpiece wall.”)

·        Line 335 – “…analyzed using specific software installed on a laptop.” – authors should elaborate.

·        Table 4. – in column 4 “observed surface” authors decided on writing: no vibration/light vibration/vibration.  Such descriptions seem rather vague, there is no mention on any criterium. In addition, there are obviously no “vibrations” to be “observed on the surface”

·        Conducting each of the experimental tests only once due to “economic and financial reasons” is not ideal. Perhaps authors could limit turning length (of which there is no mention in the paper) and conduct more tests?

·        There is no mention in the paper on tool wear, which influences cutting force components. Did authors took it into consideration? It should be clearly addressed in the paper.

Comments on the Quality of English Language

The language of the paper is mainly ok, although there are quite a few language errors and typos that need to be addressed, i.e. line 23 “…parts during due to their…”, line 28 “…higher equal…”, line 399 “…it can conclude…” etc.

Reviewer 2 Report

Comments and Suggestions for Authors

The mansucript is good, but needs a few improvements before the acceptance. In the file attached are my comments.

Round 2

Reviewer 2 Report

Comments and Suggestions for Authors

accepted.